# Induction of Chicken Host Defense Peptides within Disease-Resistant and -Susceptible Lines

**DOI:** 10.3390/genes11101195

**Published:** 2020-10-14

**Authors:** Hyun-Jun Jang, Melissa Monson, Michael Kaiser, Susan J Lamont

**Affiliations:** 1Department of Animal Science, Iowa State University, Ames, IA 50011, USA; 316wkd@hanmail.net (H.-J.J.); msmonson@iastate.edu (M.M.); mgkaiser@iastate.edu (M.K.); 2Department of Animal Biotechnology, Jeonbuk National University, Jeonju 54896, Korea; 3Department of Research and Development, Center for Industrialization of Agricultural and Livestock Microorganisms, Jeongup 56212, Korea

**Keywords:** host defense peptides, gene expression, immune responses, chicken

## Abstract

Host defense peptides (HDPs) are multifunctional immune molecules that respond to bacterial and viral pathogens. In the present study, bone marrow-derived cells (BMCs) and chicken embryonic fibroblasts (CEFs) were cultured from a Leghorn line (Ghs6) and Fayoumi line (M15.2), which are inbred chicken lines relatively susceptible and resistant to various diseases, respectively. The cells were treated by lipopolysaccharide (LPS) or polyinosinic-polycytidylic acid (poly(I:C)) and, subsequently, mRNA expression of 20 chicken HDPs was analyzed before and after the stimulation. At homeostasis, many genes differed between the chicken lines, with the Fayoumi line having significantly higher expression (*p* < 0.05) than the Leghorn line: *AvBD1*, *2*, *3*, 4, *6*, and *7* in BMCs; *CATH1*, *CATH3*, and *GNLY* in CEFs; and *AvDB5*, *8*, *9*, *10*, *11*, *12*, *13* in both BMCs and CEFs. After LPS treatment, the expression of *AvBD1*, *2*, *3*, *4*, *5*, *9*, *12*, *CATH1*, and *CATHB1* was significantly upregulated in BMCs, but no genes changed expression in CEFs. After poly(I:C) treatment, *AvBD2*, *11*, *12*, *13*, *CATHB1* and *LEAP2* increased in both cell types; *CATH2* only increased in BMCs; and *AvBD3*, *6*, *9*, *14*, *CATH1*, *CATH3*, and *GNLY* only increased in CEFs. In addition, *AvBD7*, *AvBD14*, *CATH1*, *CATH2*, *GNLY*, and *LEAP2* showed line-specific expression dependent upon cell type (BMC and CEF) and stimulant (LPS and poly(I:C)). The characterization of mRNA expression patterns of chicken HDPs in the present study suggests that their functions may be associated with multiple types of disease resistance in chickens.

## 1. Introduction

Thousands of host defense peptides (HDPs), also known as antimicrobial peptides, have been identified in various organisms including bacteria, fungi, plants and animals [1,2]. HDPs exhibit antimicrobial activities against bacteria, fungi and viruses by damaging their cell membranes or organelles, or by disrupting their physiological activities [2,3,4]. In addition, HDPs of humans, animals and plants may act as signaling molecules in innate immune defense, chemokine induction, chemotaxis, inflammation and wound healing. [5,6,7,8].

The primary avian HDPs studied to date are the defensins, cathelicidins, liver-expressed antimicrobial peptide 2 (*LEAP2*), and NK-lysin [9,10,11,12,13,14]. Of the three main defensins (α-, β-, and θ-defensins), birds have only β-defensins, which are regarded as the ancestor of the others [15,16,17]. Chickens have 14 avian β-defensin (*AvBD1-14*) and 4 ovodefensin (*OvoDA1*, *1–2*, *3*, and *B1*) genes clustered on chromosome 3 [18,19]. Transcripts of AvBDs were first found in leukocytes and bone marrow [20,21] and they have since been reported in immune-related tissues, the respiratory tract, and the reproductive tract [9,10] while ovodefensin was only expressed in egg white and the oviduct [19]. Chickens have four cathelicidin (*CATH1*, *CATH2*, *CATH3*, and *CATHB1* also known as fowlicidin-1, -2, -3 and chCATH-B1) genes clustered on chromosome 2. The avian cathelicidins have a conserved signal peptide and cathelin domain similar to mammalian cathelicidins [22,23]. *CATH1*, *2*, and *3* are expressed in various tissues including the respiratory tract, gastrointestinal tract and multiple lymphoid organs, while *CATHB1* shows specifically higher expression in bursa compared to other tissues [10,24,25]. The *LEAP2* gene is on chicken chromosome 13. Chicken *LEAP2* has approximately 60% homology to mammalian *LEAP2* and is expressed in liver, intestine, gall bladder, kidney, and multiple reproductive organs [26,27]. Chicken NK-lysin (*GNLY*) has been recently reported to have antimicrobial and anticancer activities. Additionally, *GNLY* is overexpressed in immune-related organs and the gastrointestinal tract, such as spleen and duodenal loop in chickens [10,11].

The Leghorn line Ghs6 was highly inbred from a cross of Leghorn lines; the Leghorn breed originated in Europe and provided the foundation stock for many commercial layer lines [28,29,30]. The Fayoumi line M15.2 was segregated from the Egyptian Fayoumi breed, has been highly inbred, and has large genetic distance from broiler and Leghorn lines [29,31]. From past comparisons between inbred Fayoumi and Leghorn chickens, the Fayoumi line has shown more resistance against various pathogens and stressful environments, suggesting superior immune responses [29,30,32,33]. In previous studies, Fayoumis have been demonstrated to have different expression of several HDPs [34] and association of genetic variants with disease response [35]. In addition, chicken bone marrow has been regarded as a secondary lymphoid organ interacting with thymus and bursa of Fabricius (bursa) [36,37,38].

The immune response of bone marrow-derived dendritic cells can be activated by various stimulating conditions such as lipopolysaccharide (LPS) and/or heat stress [33]. Fibroblasts have been early considered as a non-immune type cell maintaining the structural integrity of connective tissue [39,40,41,42]. Additionally, avian immune responses have been studied by comparing between fibroblasts and immune typed cells [43,44,45]. Collectively, the comparisons using these cell types can provide insight on systemic immune responses between and within chicken lines.

LPS and polyinosinic-polycytidylic acid (poly(I:C)) are commonly used to mimic bacteria- and virus-mediated immune responses, respectively. LPS originates from the outer membrane of gram-negative bacteria and poly(I:C) is a synthetic molecule that has a double-stranded RNA structure like an RNA-viral genome [46,47,48]. TLR4 on cell membrane and TLR3 located in endosome vesicles are activated by LPS and poly(I:C), respectively. After the TLRs are bound, then the related signaling pathways induce host immune responses [49,50,51,52,53]. Chicken cells respond to LPS and poly(I:C) in a similar manner that is generally comparable to other species. It has been reported that various cytokines and immune-related genes were induced in several chicken cell types by LPS and poly(I:C) [33,54,55,56,57]. However, stimulation of HDPs by LPS or poly(I:C) is not yet well characterized in chicken.

In this study, we analyzed the in vitro expression of the major chicken HDPs (excluding egg-specific HDPs) after induction by LPS and poly(I:C) of fibroblast and bone marrow-derived cells from inbred Fayoumi and Leghorn chickens. Our study using induction by bacterial and viral components may contribute to understanding expression patterns of HDPs against various pathogens in chickens.

## 2. Materials and Methods

### 2.1. Ethical Statement

The Institutional Animal Care and Use Committee at Iowa State University (ISU) approved all animal procedures (IACUC-19-287). All methods were performed in accordance with the relevant guidelines and regulations outlined in this protocol. Fertilized eggs from the inbred Leghorn line Ghs6 and the Fayoumi line M15.2 were incubated in three batches to obtain in total 90 5-day-old embryos per line and 90 day-of hatch chicks per line.

### 2.2. Cell Culture

#### 2.2.1. Chicken Embryonic Fibroblast (CEF) Culture

The head, internal organs, limbs, and tail were removed from 5-day-old embryos. The remaining embryonic bodies were rinsed with 1 × phosphate buffered saline (PBS) (Gibco, Thermo Fisher Scientific, Waltham, MA, USA). Thirty samples per line were pooled within line (Fayoumi or Leghorn) for each of the 3 batches, resulting in a total of 6 pooled samples of CEF. Each pooled sample was then dissociated using 0.05% Trypsin EDTA (TE) (Gibco) at 39 °C for 10 min. Subsequently, 4 × 10^5^ of the dissociated cells were cultured per well in multiple 6-well culture plates with 2 mL of Dulbecco’s Modified Eagle’s Medium (DMEM) (Gibco) with 10% fetal bovine serum (FBS) (Hyclone, Logan, UT, USA) and 1% antibiotics (1× Antibiotic-Antimycotic, Gibco) at 39 °C with 5% CO2 until passage 3 (P3). For passaging each cell batch, CEFs from a single well were split into 3 wells in every subculture. CEFs were frozen in liquid nitrogen at P3 until treatments were performed. Before treatments, equal numbers of CEFs from all three batches were mixed, and then 5 × 10^5^ of the mixed batch cells were cultured until passage 5 (P5). The cell viability, determined by trypan blue staining, was over 95% in every subculture.

#### 2.2.2. Bone Marrow-Derived Cell (BMC) Culture

For each batch, both thighbones were collected from 30 day-of-hatch chicks per line. Each thighbone was split and bone marrow was scraped from the split thighbone using a sharp blade. The bone marrows were rinsed with 1 × PBS (Gibco), then pooled within line within each of the 3 batches, resulting in a total of 6 pooled bone marrow samples. Each pooled sample was then dissociated using 0.05% TE (Gibco) at 39 °C for 20 min. Subsequently, 3 × 10^5^ of the dissociated cells were cultured per well in multiple 6-well culture plates with 2 mL of DMEM (Gibco) with 10% FBS (Hyclone), 5% chicken serum (Gibco) and 1% antibiotics (1 × Antibiotic-Antimycotic, Gibco) at 39 °C with 5% CO2 until passage 2 (P2). For passaging each cell batch, BMCs from a single well were split into 4 wells in every subculture. BMCs were frozen in liquid nitrogen at P2 until treatments were performed. Before treatments, equal of BMCs from all three batches were mixed, and then 5 × 10^5^ of the mixed batch cells were cultured until P3. The cell viability, determined by trypan blue staining, was over 95% in every subculture.

### 2.3. Immune Stimulation

#### 2.3.1. Lipopolysaccharide (LPS) Treatment

P3 BMCs and P5 CEFs were prepared to 70~80% confluency in 6-well culture plates with 2 wells for each experimental group before exposure to LPS treatment. To mimic an extracellular bacterial infection, LPS (Salmonella enterica serotype typhimurium, L6143, Sigma-Aldrich, St. Louis, MO, USA) was added to the prepared cells at concentrations of 100 or 200 ng/mL at 1, 3, 6, 12, 24, and 48 h before sampling the cultures. Within each experimental group, the cells in 2 wells were pooled during sampling to generate sufficient material for subsequent assays. All LPS-treated groups were simultaneously sampled at the end of the experiment. A non-LPS-treated group was also sampled before starting the treatment and at the end of the experiment to assess the change of HDP expression during cell incubation without stimulant. The treatment was independently performed using each cell batch, providing three experimental replicates of each LPS- or non-treated group. In total, 168 samples (2 chicken lines × 2 cell types × 2 concentrations of LPS × 6 exposure times × 3 LPS-stimulated replicates, plus 2 chicken lines × 2 cell types × 2 exposure times × 3 non-stimulated replicates) were individually snap frozen using liquid nitrogen and stored at −80 °C until use.

#### 2.3.2. Polyinosinic: Polycytidylic Acid (Poly(I:C)) Treatment

P3 BMCs and P5 CEFs were prepared to 70~80% confluency in 6-well culture plates with 2 wells for each experimental group before exposure to poly(I:C) treatment. To mimic an intracellular viral infection, poly(I:C) (Poly(IC) HMW, InvivoGen, San Diego, CA, USA) was transfected into the prepared cells at concentrations of 10 and 50 ng/mL using lipofectamine (Lipofectamine 3000 Transfection Reagent, Invitrogen, Thermo Fisher Scientific). Within each experimental group, the cells in 2 wells were pooled during sampling to generate sufficient material for subsequent assays. Each poly(I:C)-treated group was sampled at 1, 3, 6, 12, and 24 h after the transfection. Non-poly(I:C)-treated controls were also sampled at 1, 3, 6, 12, and 24 h after lipofectamine treatment without poly(I:C). In addition, no-treatment controls (without lipofectamine) were sampled before the treatment and at the same exposure times as the treatment groups (1, 3, 6, 12, and 24 h) to determine the HDP expression during cell incubation. The treatment was independently performed using each cell batch, providing three experimental replicates of each poly(I:C)-, non-poly(I:C)- or non-treated group. In total, 252 samples (2 chicken lines × 2 cell types × 2 concentrations of poly(I:C) × 5 exposure times × 3 poly(I:C)-stimulated replicates, plus 2 chicken lines × 2 cell types × 5 exposure times × 3 non-poly(I:C)-lipofectamine only replicates, plus 2 chicken lines × 2 cell types × 6 sampling times (including 0 h) × 3 non-stimulated replicates) were rapidly frozen using liquid nitrogen and stored at −80 °C until further use.

### 2.4. Analysis of Gene Expression

#### 2.4.1. Primer Design

Primers for 20 targeted HDP genes and 3 reference genes were designed using Primer-BLAST (https://www.ncbi.nlm.nih.gov/tools/primer-blast/index.cgi?LINK_LOC=BlastHome) (Table 1). The amplicon identity was confirmed to match the intended target gene by Sanger sequencing for all 23 primer sets.

#### 2.4.2. Expression Analysis of Host Defense Peptide Genes

Total RNA was isolated from the samples using the RNAqueous Total RNA Isolation Kit (Ambion, Thermo Fisher Scientific) according to the manufacturer’s protocol. RNA quantity was assessed on a NanoDrop ND-1000 UV-vis spectrophotometer (Thermo Fisher Scientific). cDNA was synthesized from 50 ng of each RNA sample using Reverse Transcription Master Mix (Fluidigm, South San Franciso, CA, USA) and it was subsequently preamplified for 16 cycles using Preamp Master Mix (Fluidigm) following manufacturer’s protocols. Each sample was assayed in duplicate using a Fluidigm 192.24 Integrated Fluidic Circuit (IFC). Real-time PCR was performed on the Biomark HD (Fluidigm) with five independent IFCs (IFC1, 2, 3, 4, and 5). IFC1 included LPS-treated BMCs and their non-treated controls from Fayoumi and Leghorn from all three experimental replicates (Figure 1). IFC2 included LPS-treated CEFs and their non-treated controls in Fayoumi and Leghorn from all three experimental replicates (Figure 1). IFC3, 4, and 5 each included one independent experimental replicate of poly(I:C)-treated BMCs and CEFs and their non-poly(I:C)-treated and non-treated controls in Fayoumi and Leghorn (Figure 1). The data were analyzed using the Fluidigm Real-Time PCR Analysis software. The technical replicates of the same sample per primer set had overlapping major peaks in their melting curves, confirming that a single product was amplified for each primer pair. Gene expression was compared between chicken lines, between LPS-treated groups and non-treated controls (from 48 h after incubation without LPS), or between poly(I:C)-treated groups and non-poly(I:C)-treated controls (with lipofectamine), all using the 2^(−ΔΔCt)^ method [58] in Excel. The relative expression of the HDPs between lines or treatments was visualized as Log2 fold changes (Log2FC) using the gplots package in R (https://ggplot2.tidyverse.org).

#### 2.4.3. Data Imputation and Reference Gene Selection

In this analysis, 35% cycle thresholds (Cts) values of HDPs were out of the range of the Biomark Dynamic assay. Hypothesizing that the out-of-range Cts were lowly expressed genes, these missing Cts were tested with imputation of 31, 50, and 100 Cts, all of which are below the detection range of 1 to 30 Cts. Missing Ct values were also imputed to 1 Ct (the minimum Ct), which is the maximum detection value. From principal component analysis (PCA), the imputations of 31, 50, and 100 Cts were similarly clustered among experimental groups while the imputation of 1 Ct was differently clustered (Appendix A). Collectively, the results suggested that the missing values occurred by exceeding the maximum Cts, indicating either no or low gene expression. Thus, 31 Cts was imputed to the missing Ct values of HDPs for downstream analysis, understanding that this may have been an overestimate of expression for some samples.

To select between the tested reference genes (*H6PD*, *GAPDH*, and *28S*), Ct variation in the reference genes due to experimental variables was statistically tested for IFC1 and 2, or for IFC3, 4, and 5 (Figure 1). Linear regression models were fit with Ct ~ time + line + LPS concentration + IFC/cell type (combined effect of IFC and cell type) for IFC1 and 2 and Ct ~ time + line + Poly(I:C) concentration + cell type + IFC for IFC 3, 4, and 5. These models revealed that the Cts of *GAPDH* showed no significant difference for most variables, except the combined effect of IFC and cell type in IFC 1 and IFC 2 (Figure 1), which were not statistically compared in the downstream analysis. Therefore, *GAPDH* was selected as the reference gene to calculate delta Ct (dCt) values.

### 2.5. Statistical Analysis

Within each cell type, multiple linear regression models were fit to the dCts using JMP 14.3.0 (SAS Institute Inc. NC, USA) to identify the effects of LPS treatment, poly(I:C) treatment or homeostatic differences between lines without stimulation of the cells. Because the different concentrations of LPS or poly (I:C) did not significantly affect gene expression of most HDPs (*p* > 0.05, data not shown), the following statistical analyses investigated the overall effects of treatment, combining the data from both concentrations. Post-hoc testing of significant interactions (*p* < 0.05) in these models was performed using the Tukey HSD test in JMP.

The LPS-treated BMCs and their non-treated controls (IFC1) or the LPS-treated CEFs and their non-treated controls (IFC2) were separately fit to the linear regression model of dCt ~ Line + Treatment + Time[Treatment] + Line*Treatment + Line*Time[Treatment] + Replicate^&Random^. In the model, line included Fayoumi and Leghorn. Treatment was categorized into LPS-treated and non-treated groups. Time was nested within treatment and included 0, 1, 3, 6, 12, 24, and 48 h before the experiment termination. Replicate included the 3 experimental replicates as a random effect.

To investigate homeostatic differences between the chicken lines within each cell type, the non-poly(I:C)-treated and non-treated controls from IFC3, 4 and 5 were subset by cell type (BMC and CEF); these subsets were separately fit with the linear regression model of dCt ~ Line + Lipofection + Time + Replicate^&Random^. In the model, line included Fayoumi and Leghorn. Lipofection included the non-poly(I:C) but lipofectamine-treated and no-treatment control groups. Time included 0, 1, 3, 6, 12, and 24 h after the experiment initiation. Replicate included the effect of the IFCs and the 3 experimental replicates as a random effect.

The poly(I:C)-treated BMCs and their non-poly(I:C)-treated controls (IFC3, 4 and 5) or the poly(I:C)-treated CEFs and their non-poly(I:C)-treated controls (IFC3, 4 and 5) were separately fit to the linear regression model of dCt ~ Line + Treatment + Time[Treatment] + Line*Treatment + Line*Time[Treatment] + Replicate^&Random^. In the model, line included Fayoumi and Leghorn. Treatment was categorized into poly(I:C)-treated and non-poly(I:C)-treated groups (with lipofectamine only). Time included 0, 1, 3, 6, 12, and 24 h after the experiment initiation. Replicate included the effect of the IFCs and the 3 experimental replicates as a random effect.

## 3. Results

### 3.1. Line Differences in Expression of HDPs in Homeostatic State

From the regression model on control (non-poly(I:C)-treated and non-treated) groups from the poly(I:C) experimental sets (IFC3, 4, and 5), homeostatic HDP expression levels were mainly affected by line, although a few HDP expression levels were affected by time and lipofectamine (Appendix A). All HDPs with a significant difference between lines (*p* < 0.05) showed higher expression in Fayoumi compared to Leghorn (Figure 2A). Among them, *AvBD5*, *8*, *9*, *10*, *11*, *12*, *13*, and *CATHB1* were higher expressed in both BMCs and CEFs while *AvBD1*, *2*, *3*, *4*, *6*, and *7* exhibited specifically higher expression in BMCs and *CATH1*, *CATH3*, and *GNLY* exhibited specifically higher expression in CEFs (Figure 2). Our data indicate that most HDPs are more highly expressed in Fayoumi than Leghorn, and BMCs and CEFs showed different expression patterns of HDPs at homeostatic levels.

### 3.2. HDPs Responded to LPS and Poly(I:C)

Regression models were also used to investigate the effect of treatment and revealed differences in which HDPs respond significantly (*p* < 0.05) to LPS or poly(I:C) in BMCs or CEFs (Appendix A and Figure 3A). After LPS treatment, the expression of *AvBD1*, *2*, *3*, *4*, *5*, *9*, *12*, *CATH1*, and *CATHB1* significantly increased in BMCs while no HDP expression significantly changed in CEFs (Figure 3A). After poly(I:C) treatment, *AvBD2*, *11*, *12*, *13*, *CATH2*, *CATHB1*, and *LEAP2* significantly increased in BMCs and *AvBD2*, *3*, *6*, *9*, *11*, *12*, *13*, *14*, *CATH1*, *CATH3*, *CATHB1*, *GNLY*, and *LEAP2* increased in CEFs (Figure 3A). Expression of 6 of these HDPs additionally had a significant interaction between treatment and line factors (Appendix A) and were subsequently used for a post-hoc Tukey test (*p* < 0.05) (Figure 3B,C). From the post-hoc Tukey test, *AvBD2* (Leghorn, *p* < 0.001; Fayoumi, *p* < 0.05) and *CATHB1* (Leghorn, *p* < 0.01; Fayoumi, *p* < 0.001) expression significantly increased in BMCs of both lines while the expression of *AvBD3* (*p* < 0.05), *AvBD5* (*p* < 0.05), and *CATH3* (*p* < 0.05) only increased in Leghorn BMCs after LPS treatment (Figure 3B). Although *AvBD3* and *AvBD5* only responded to LPS in Leghorn BMCs, their expression when stimulated still remained lower than the Fayoumi BMCs without LPS. Similarly, while LPS increased *AvBD2* expression in both lines, the Fayoumi BMCs had greater expression than the Leghorn BMCs under both LPS- and non-treated conditions. *CATHB1* had higher homeostatic levels in the Leghorn BMCs, but after LPS treatment, the Fayoumi BMCs had higher expression. *AvBD7* expression significantly increased in Fayoumi CEFs after poly(I:C) treatment (*p* < 0.01), but there was no response in the Leghorn line (Figure 3C). There were no significant interactions between line and treatment in CEFs after LPS treatment or in BMCs after poly(I:C) (Appendix A). Collectively, our data indicate that under our treatment conditions, poly(I:C) stimulated various HDPs in BMCs and CEFs while LPS stimulated them in the only BMCs. Especially, *AvBD2*, *AvBD3*, *AvBD5*, *CATH3*, *CATHB1* showed an interaction between line and LPS, while *AvBD7* responded to poly(I:C) in a line-dependent manner.

### 3.3. Line-Dependent Expression of HDPs Impacted by LPS and Poly(I:C) Treatments

To determine whether the line differences observed under homeostatic conditions were affected by adding LPS- and/or poly(I:C)-stimulated groups, HDPs were compared between lines in BMCs or CEFs using the regression models for LPS- and non-treated groups (IFC 1 and 2) or poly(I:C)- and non-poly(I:C)-treated groups (IFC 3, 4, and 5). Comparing Fayoumi to Leghorn in BMCs with poly(I:C) treatment (poly(I:C) + non-poly(I:C)-treated), Fayoumi BMCs had higher expression of *AvBD1*, *2*, *3*, *4*, *5*, *6*, *7*, *8*, *9*, *10*, *11*, *12*, *13*, and *CATHB1*, maintaining the difference between lines observed in the homeostatic state (Figure 4A,B). Additionally, *AvBD14*, *CATH2*, and *LEAP2* were significantly higher in Fayoumi than Leghorn BMCs with poly(I:C) treatment, but not at homeostasis. Similarly, *AvBD1*, *2*, *3*, *4*, *5*, *6*, *7*, *9*, and *10* were expressed at greater levels in Fayoumi than Leghorn BMCs with LPS treatment (LPS + non-treated), maintaining line-dependent expression patterns from the homeostatic state. However, incorporating the LPS treatment (LPS + non-treated) did change some expression patterns in BMCs; *AvBD14* became significantly higher expressed in Fayoumi than Leghorn BMCs, while *CATH1*, *CATH2*, *CATHB1*, *GNLY*, and *LEAP2* were greater in Leghorn than Fayoumi BMCs only with LPS treatment (Figure 4A,B). Comparing Fayoumi to Leghorn in CEFs with poly(I:C) treatment, Fayoumi CEFs had significantly higher expression of *AvBD5*, *8*, *9*, *10*, *11*, *12*, *13*, *CATH1*, *CATH3*, *CATHB1*, and *GNLY* than Leghorn CEFs, which are the same line-dependent expression patterns observed in the homeostatic state (Figure 4A,C). Addition of the LPS treatment (LPS + non-treated) maintained less of the homeostatic line differences in the CEFs, but *AvBD5*, *AvBD12*, *AvBD13* and *CATHB1* were more highly expressed in Fayoumi than Leghorn CEFs both with LPS treatment and at homeostasis. Additionally, *AvBD7*, *CATH2*, and *LEAP2* had significantly greater expression in Fayoumi than Leghorn CEFs only with poly(I:C) treatment (Figure 4A,C). These results suggest that *AvBD7*, *14*, *CATH1*, *CATH2*, *CATHB1*, *GNLY*, and *LEAP2* had line differences in expression in a stimulant-dependent (LPS and poly(I:C)) manner.

## 4. Discussion

The inbred Fayoumi line included in this study has been shown to be relatively more resistant against various pathogens and environmental stresses than the inbred Leghorn line or a broiler line, including the Fayoumi line activating immune-related signaling pathways against viral infection and LPS treatment with/without heat stress [32,59,60,61]. In this study, we investigated the expression of 20 chicken HDPs before and after induction by LPS and poly(I:C) in embryonic fibroblasts and bone marrow-derived cells from the inbred Fayoumi M15.2 and Leghorn Ghs6 chicken lines.

From comparisons between the two lines under homeostatic conditions, all HDPs except *AvBD14*, *CATH2*, and *LEAP2* showed higher expression in Fayoumi than Leghorn. HDPs are well known as a critical component of immune responses that inhibit microbes such as bacteria and viruses and modulate the host immune system [5,8,62]. Many studies have also suggested an association between HDP expression and various infectious diseases [63,64,65], though the direct evidence for a correlation between a HDP and disease resistance is still limited in livestock. Thus, our results suggested that the higher expression of HDPs in the Fayoumi line at homeostasis could be associated with its innate disease resistance.

LPS and poly(I:C) are well known as agonists of Toll-like receptors (TLRs), which recognize microbial pathogens such as bacteria and viruses, triggering host innate immune responses. LPS and poly(I:C) are respectively recognized by TLR4 on the host cell membrane and TLR3, which resides in endosome vesicles [52,53]. After that, the host immune system activates various cytokines and chemokines through serial signaling pathways. Ultimately, the chain of such immune signaling pathways activates HDP expression, although the regulatory mechanisms of HDP expression are still unclear in chickens [49,50,51]. In this study, *AvBDs* generally maintained higher expression in the Fayoumi line than Leghorn line after LPS and poly(I:C) treatments, while the expression patterns of *CATH1*, *CATH2*, *CATHB1*, *GNLY*, and *LEAP2* between the two lines varied according to the stimulant (LPS or poly(I:C)). Previous studies have reported that 14 AvBD genes are densely clustered on chicken chromosome 3 and their functional domains are highly conserved [16,26,66,67,68]. Another study suggested that there are interactions between AvBDs and other antimicrobial proteins or among AvBD family proteins [45]. Therefore, our results suggest that the genes of the AvBD family could be systemically upregulated after immune stimulation to cooperatively confer disease resistance to the Fayoumi line.

In addition to the AvBDs, the CATH family genes, *GNLY*, and *LEAP2* showed variable expression patterns across lines, cell types, or stimulant treatments. Unlike human and mouse, which have a single cathelicidin on their genomes, chicken has four cathelicidins (*CATH1*, *CATH2*, *CATH3*, and *CATHB1*) [23,25]. CATH1, 2, and 3 share a typical cathelin-like domain with mammalian cathelicidins, while CATHB1 has an uncharacteristic cathelin-like domain. CATHB1 antimicrobial activity is known to be relatively weaker than the other cathelicidins [10,23,24,25]. CATH1 and CATH3 have a high similarity of amino acid sequences compared to others (>90%), suggesting the same gene clade of CATH gene family. In addition, *CATH1*, 2, and 3 can be highly expressed in chicken bone marrow [10], while *CATHB1* was characteristically expressed in the bursa [10,25]. Chicken *LEAP2* can not only be widely expressed in the liver, intestine, gall bladder, kidney, and multiple reproductive organs but can also increase during embryo development [23,26,27,69]. Chicken *GNLY* was shown to be characteristically overexpressed in spleen and the duodenal loop compared to AvBDs and CATHs [10]. Collectively, these studies suggest that these HDPs can have independent or cooperative roles in various tissues, although their specific functions are still unclear.

## 5. Conclusions

In conclusion, we analyzed the expression of the major chicken HDPs and confirmed the higher expression of most HDPs in the disease-resistant Fayoumi line compared to the disease-susceptible Leghorn line at homeostasis and in an LPS- or poly(I:C)-stimulated state. Thus, our results suggest that HDP functions may be associated with disease resistance mechanisms in chicken. Functions and regulatory mechanisms of these HDPs should be further studied to understand the relationship between specific pathogen resistance and HDPs in these inbred chicken lines and commercial populations.

## Figures and Tables

**Figure 1 genes-11-01195-f001:**
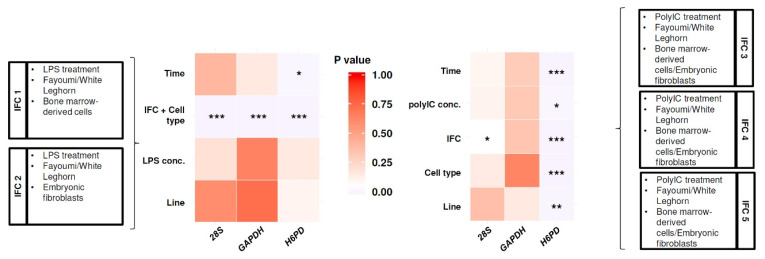
Statistical analysis for impact of experimental variables on Ct values of reference genes among Fluidigm 192.24 Integrated Fluidic Circuits (IFCs). *p*-values for each experimental variable and loaded sample information for each IFC are shown. *, *p* < 0.05; **, *p* < 0.01; ***, *p* < 0.001.

**Figure 2 genes-11-01195-f002:**
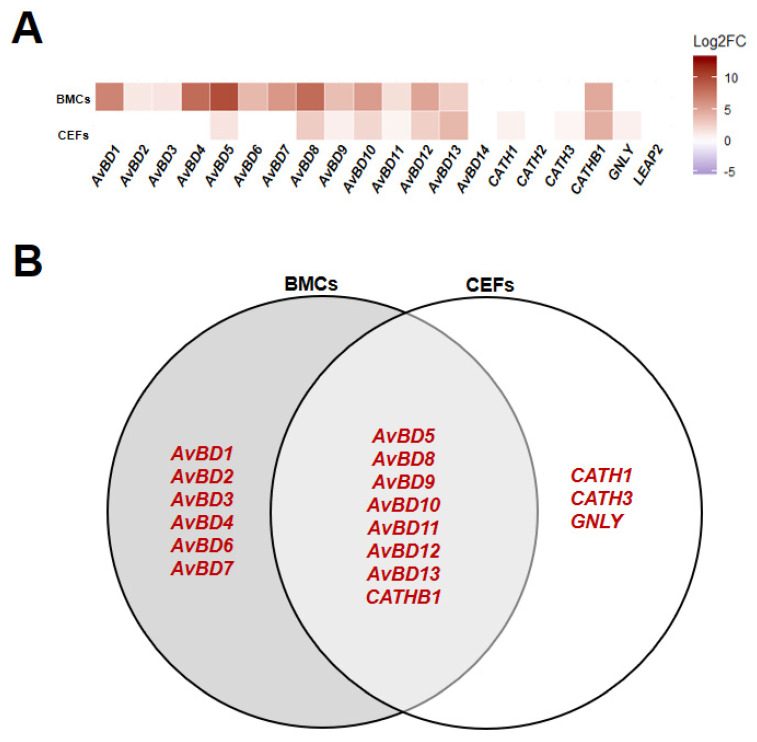
Expression patterns of host defense peptides (HDPs) in homeostatic state comparing Fayoumi to Leghorn within each cell type. Relative expression state (Log_2_ fold change (Log2FC)) of HDPs in bone marrow-derived cells (BMCs) and chicken embryonic fibroblasts (CEFs) comparing Fayoumi to Leghorn (**A**) and Venn diagram of HDPs that were significantly differentially expressed (*p* < 0.05) in BMCs and CEFs (**B**). Red font indicates higher expression in Fayoumi compared to Leghorn.

**Figure 3 genes-11-01195-f003:**
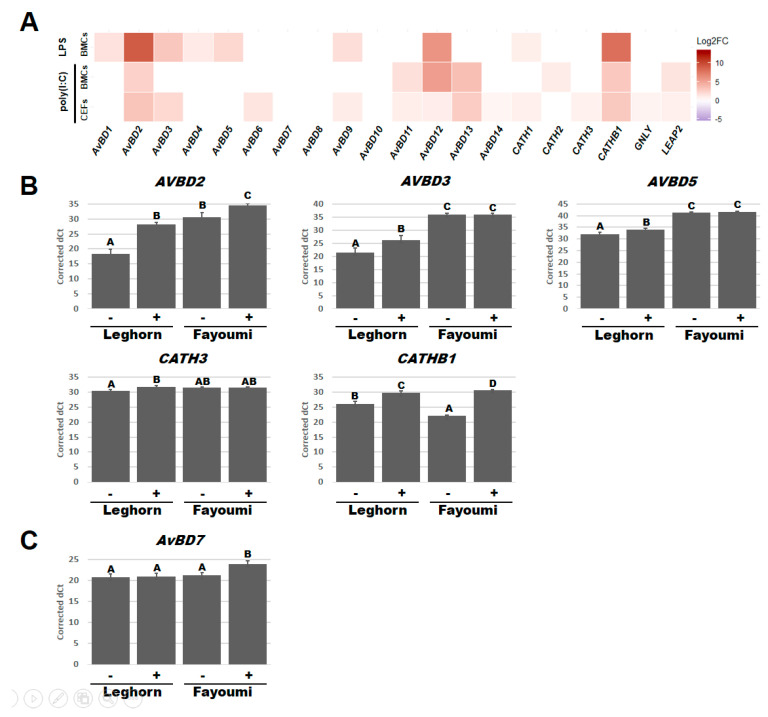
Host defense peptides (HDPs) that responded to LPS or poly(I:C) treatment. Heatmap for Log_2_ fold change (Log2FC) of each HDPs comparing the LPS- or poly(I:C)-treated cells to the non-treated or non-poly(I:C)-treated cells in BMCs or CEFs (**A**). Bar graph of corrected dCts for HDPs with a significant interaction between line and LPS in BMCs (**B**) and between line and poly(I:C) in CEFs (**C**). Corrected dCts were calculated by 30 (maximum detected dCt)—lsmean of the targeted gene dCt and LSmeans adjusted according to each linear model. BMCs, bone marrow-derived cells; CEFs, chicken embryonic fibroblasts; -, with treatment; +, without treatment. Statistical significance was indicated as a different character above each bar in graphs.

**Figure 4 genes-11-01195-f004:**
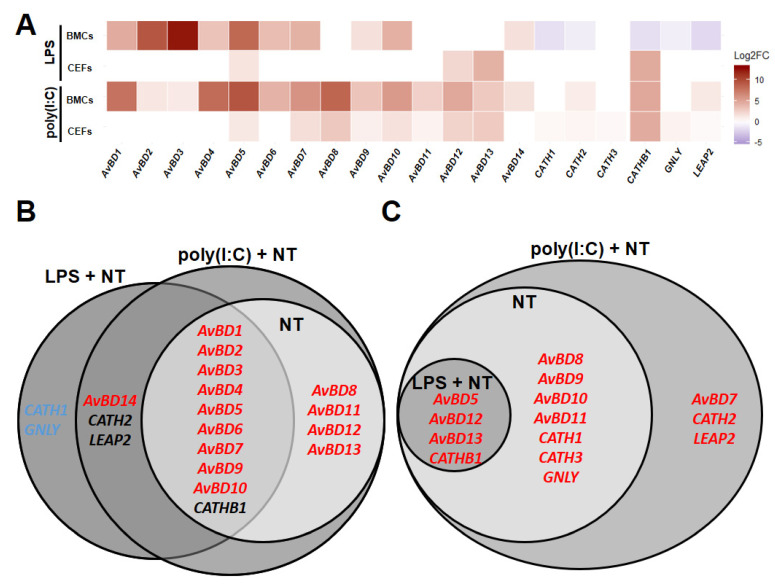
Line-dependent expression of host defense peptides (HDPs) with LPS or poly(I:C) treatment. Relative expression (Log_2_ fold change (Log2FC)) of HDPs comparing Fayoumi to Leghorn in LPS- + non-treated (LPS + NT) or poly(I:C)- + non-poly(I:C)-treated (poly(I:C) + NT) BMCs or CEFs (**A**), Venn diagram for significantly differentially expressed (*p* < 0.05) HDP in BMCs comparing Fayoumi to Leghorn in non-treated (NT), LPS + NT or poly(I:C) + NT. (**B**) Venn diagram for significantly differentially expressed (*p* < 0.05) HDP in CEFs comparing Fayoumi to Leghorn in NT, LPS + NT or poly(I:C) + NT. (**C**) Red font color indicates significantly higher expression in Fayoumi compared to Leghorn. Blue font color indicates significantly higher expression in Leghorn compared to Fayoumi. Black font color indicates higher expression in Fayoumi with poly(I:C) and in Leghorn with LPS. BMCs, bone marrow-derived cells; CEFs, chicken embryonic fibroblasts.

**Table 1 genes-11-01195-t001:** Primer information for host defense peptide genes and reference genes.

Gene	Entrez ID	Forward Primer (5’–3’)	Reverse Primer (5’–3’)	Amplicon Length (bp)
*AvBD1*	395841	ACAGACGTAAACCATGCGGA	GACTTCCTTCCTAGAGCCTGG	99
*AvBD2*	395840	GGGTGTCCCAGCCATCTAAT	TCCAAGGCCATTTGCAGCAG	70
*AvBD3*	395363	CTCTTGTTTCTCCAGGGTGCT	GCTCCCAACACGACAGAATC	75
*AvBD4*	414342	CAGTCTGCCTTCTGCCATGA	GGTCCCGCGATATCCACATT	124
*AvBD5*	414340	GTCATGTCCTCCAGGGATCG	CGTGAAGGGACATCAGAGGC	101
*AvBD6*	407776	GGGTTGGATCATGTGGCAGT	AGTGCCAGAGAGGCCATTTG	120
*AvBD7*	407777	GGGATCTGTCGAAGGCCATA	TTCCCAGAAGTCAGGGAGGT	105
*AvBD8*	414875	CTTGGCCGTTCTCCTCACTG	ACTGTGCCTCGTTGTTAGGT	71
*AvBD9*	414343	ACACCGTCAGGCATCTTCAC	GTCTTCTTGGCTGTAAGCTGG	129
*AvBD10*	414341	TCAGGGGAATTTCTGCCGTG	CTTACTGCGCCGGAATCTTG	107
*AvBD11*	414876	ATGCTCTTGGCGTCAGAAAAC	GGAGATACGCAATGGCCCC	87
*AvBD12*	414339	CTGCTCGCTCACGGAAGCA	TATTCCCCAGGGTTGCAGTTC	89
*AvBD13*	414877	AGCTGCTCTTTGCCATCGTT	CAGTGGCCATGGTTGTTCCT	97
*AvBD14*	100858701	GGCGACACGACAATGTCAAC	TTGCCCTTCATCTTCCGACA	119
*CATH1*	414337	GACTCCATGGCTGACCCTGT	ATCGCCCGGTAGAGGTTGTA	89
*CATH2*	420407	CGACTGCGACTTCAAGGAGAA	GATCTCGGGAGTGTCCTGC	79
*CATH3*	100858343	GATGTCACCTGCGTGGACTC	TTGTAGAGGTTGATGCCCGC	107
*CATHB1*	100858412	GGTTGCTCAACCAGAGGATCT	TCCTCCACAAGGAAGCTCAC	118
*GNLY*	693257	TTCTGCGTCAGTCTGGTGAA	AGATACTCCTCTGGCGCCTC	118
*LEAP2*	414338	GTTGGAGCCTCATGTAGGGA	GAGGCCGTTCTAAGGAAGCAG	80
*28S*	112533599	GGCGAAGCCAGAGGAAACT	GACGACCGATTTGCACGTC	62
*GAPDH*	374193	GCAGCAGGAACACTATAAAGGC	TTTGCCAGAGAGGACGGC	100
*H6PD*	428188	ATGTACCGGGTGGACCACTA	AACTGACGGTTCTGATCTCGAAA	80

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
