# Peer review of "Induction of Chicken Host Defense Peptides within Disease-Resistant and -Susceptible Lines"

_genes, 2020, doi:10.3390/genes11101195_

Round 1

Reviewer 1 Report

I was wondering how the authors decided to go with 100 and 200 ng/ml of LPS for challenging the cells. It seems to be a low concentration to activate the cells ! In addition, there is no evidence showing the cells are indeed activated. They could include genes involved in LPS/TLR4 transduction pathway such as CD14, TLR4, MyD88, NF-Kappa beta, MAPK1 and MAPK2.

Author Response

Response to Reviewer 1 Comments

Point 1: I was wondering how the authors decided to go with 100 and 200 ng/ml of LPS for challenging the cells. It seems to be a low concentration to activate the cells!

Response 1: In a previously reported study, we cultured chicken bone marrow derived dendritic cells (BMDC) from both Fayoumi and Leghorn lines. The immune-related genes including CCL4, CCL5, CD40, GM-CSF, IFN-γ, IL-10, IL-12β, IL-1β, IL-6, IL-8, and iNOS were stimulated by 200 ng/ml LPS. The LPS-induced gene expression was different between the lines [1]. Thus, we observed that the LPS concentration was adequate to stimulate a response and expected that it would also be adequate to observe differentially expressing HDPs between the lines. Additionally, to potentially identify a lower threshold range of LPS for inducing HDPs, we added the lower concentration (100 ng/ml LPS) treatment.

Point 2: In addition, there is no evidence showing the cells are indeed activated. They could include genes involved in LPS/TLR4 transduction pathway such as CD14, TLR4, MyD88, NF-Kappa beta, MAPK1 and MAPK2.

Response 2

We observed the induction of several HDPs which are closely associated with LPS stimulation in chicken [2,3]. Thus, we concluded that our cells were stimulated by LPS treatments. We also cited reports of various immune responses induced by LPS and poly(I:C) treatments, in line 76-81 of the introduction.

References

  1. Van Goor, A.; Slawinska, A.; Schmidt, C.J.; Lamont, S.J. Distinct functional responses to stressors of bone marrow derived dendritic cells from diverse inbred chicken lines. Dev Comp Immunol 2016, 63, 96-110, doi:10.1016/j.dci.2016.05.016.
  2. Mohammed, E.S.I.; Isobe, N.; Yoshimura, Y. Effects of Probiotics on the Expression of Cathelicidins in Response to Stimulation by Salmonella Minnesota Lipopolysaccharides in the Proventriculus and Cecum of Broiler Chicks. J Poult Sci 2016, 53, 298-304, doi:10.2141/jpsa.0160064.
  3. Zhang, L.; Lu, L.; Li, S.; Zhang, G.; Ouyang, L.; Robinson, K.; Tang, Y.; Zhu, Q.; Li, D.; Hu, Y., et al. 1,25-Dihydroxyvitamin-D3 Induces Avian beta-Defensin Gene Expression in Chickens. PLoS One 2016, 11, e0154546, doi:10.1371/journal.pone.0154546.

Finally, I have attached the revised manuscript highlighted.

Reviewer 2 Report

This manuscript describes the expression of host defence peptides in two lines of chickens following treatment with LPS (to represent bacterial infection) and Polyinosinic:polycytidylic acid (to represent viral infection). The authors utilize a panel of 20 defence peptides to analyze expressioin levels using a Fluidigm real-time PCR. Statistical analysis of output data revealed higher expression levels of defence peptides in disease-resistant Fayoumi line.

The manuscript is clearly written, including relevant introduction and detailed explanation of methodology and rationale. The conclusions are supported by the output data.

The supplementary figures weren't available, so these haven't been considered in the manuscript review.

Some minor corrections to be addressed:

Line 66 Please use a new paragraph for “The immune response…”

Line 81 There is a heading for “Ethical Statement”, but there is no statement included.

Line 239 Figure 2 legend states that red font indicates higher expression… However, all of the HDP genes shown in panel B are red.

Figure 3 It is not clear what the A, AB, B, C, D labels above the bars mean.

Author Response

Response to Reviewer 2 Comments

 This manuscript describes the expression of host defence peptides in two lines of chickens following treatment with LPS (to represent bacterial infection) and Polyinosinic:polycytidylic acid (to represent viral infection). The authors utilize a panel of 20 defence peptides to analyze expressioin levels using a Fluidigm real-time PCR. Statistical analysis of output data revealed higher expression levels of defence peptides in disease-resistant Fayoumi line.

The manuscript is clearly written, including relevant introduction and detailed explanation of methodology and rationale. The conclusions are supported by the output data.

The supplementary figures weren't available, so these haven't been considered in the manuscript review.

Some minor corrections to be addressed:

Point 1: Line 66 Please use a new paragraph for “The immune response…”

Response 1: We made a new paragraph for it.

Point 2: Line 81 There is a heading for “Ethical Statement”, but there is no statement included.

Response 2: We added the Ethical Statement.

Point 3: Line 239 Figure 2 legend states that red font indicates higher expression… However, all of the HDP genes shown in panel B are red.

Response 3: All HDPs presented in Figure 2 were higher expression in Fayoumi compared to Leghorn. In this case, we used red color to unify color definition through all Figures.

Point 4: Figure 3 It is not clear what the A, AB, B, C, D labels above the bars mean.

Response 4: Statistical significance was indicated as different characters such as “A”, “B”, “C”, or “D” above each bar in graphs. We also added this description in the legend of Figure 3.

Finally, I have attached the revised manuscript highlighted.

Round 2

Reviewer 1 Report

I believe this manuscript needs major revision and it is not qualify for such high impact journal.